# Shapley-Based Data Valuation for Weighted $k$-Nearest Neighbors

**Guangyi Zhang**
Shenzhen Technology University
zhangguangyi@sztu.edu.cn

**Qiyu Liu**[*]
Southwest University
qyliu.cs@gmail.com

**Aristides Gionis**[*]
KTH Royal Institute of Technology and Digital Futures
argioni@kth.se

## Abstract

Data valuation quantifies the impact of individual data points on model performance, and Shapley values provide a principled approach to this important task due to their desirable axiomatic properties, albeit with high computational complexity. Recent breakthroughs have enabled fast computation of exact Shapley values for unweighted $k$-nearest neighbor ($k$NN) classifiers. However, extending this to weighted $k$NN models has remained a significant open challenge. The state-of-the-art methods either require quadratic time complexity or resort to approximation via sampling. In this paper, we show that a conceptually simple but overlooked approach — data duplication — can be applied to this problem, yielding a natural variant of weighted $k$NN-Shapley. However, a straightforward application of the data-duplication idea leads to increased data size and prohibitive computational and memory costs. We develop an efficient algorithm that avoids materializing the duplicated dataset by exploiting the structural properties of weighted $k$NN models, reducing the complexity to near-linear time in the original data size. Besides, we establish theoretical foundations for this approach through axiomatic characterization of the resulting values, and empirically validate the effectiveness and efficiency of our method.

## 1 Introduction

In the era of data-driven machine learning, understanding the value and contribution of individual data points has emerged as a critical challenge. Data valuation—the process of quantifying the impact of each training instance on model performance—plays a pivotal role in numerous applications, including identifying influential samples, detecting mislabeled data, and designing data markets [1, 2]. As machine-learning systems continue to proliferate across domains, the development of robust and scalable data-valuation methods has become increasingly essential.

Among various approaches for data valuation [3, 4, 5], the Shapley value from cooperative game theory has attracted significant attention due to its unique desirable axiomatic properties [6]. When applied to machine learning, data Shapley values provide a principled framework for distributing the model's performance among training instances, where each data point's Shapley value represents its average marginal contribution across all possible subsets of the dataset [1].

Despite its theoretical appeal and desirable properties, the exact computation of data Shapley values presents substantial challenges, as it requires enumerating and re-training over all possible subsets

---

[*]Corresponding authors

of data points—a task whose complexity grows exponentially with the dataset size. Indeed, the computation has been proven to be #**P**-hard in certain games [7]. Fortunately, a notable recent advancement in this domain by Jia et al. [8] leverages the structural properties of unweighted $k$-nearest neighbor ($k$NN) classifiers to efficiently calculate exact data Shapley values in closed form, referred to as unweighted $k$NN-SV hereafter. The $k$NN approach is a classic algorithm in machine learning, and it has been shown that running $k$NN over pre-trained embeddings can yield comparable performance with more advanced models [9].

However, a significant limitation of the approach by Jia et al. [8] lies in its restriction to unweighted $k$NN models. In practice, weighted $k$NN models, which assign different importance to neighbors based on their distances, offer greater flexibility and typically yield superior performance. Computing Shapley values for weighted $k$NN models poses unique challenges due to the lack of a closed-form solution analogous to the unweighted case. To date, the only successful attempt has been a hard-label variant of weighted $k$NN-SV [10], which invokes a sophisticated and time-consuming dynamic-programming approach that is similar in spirit to those for power indexes [11].

In this paper, we offer a novel variant of weighted $k$NN-SV, dubbed as D$k$NN-SV for duplication-based weighted $k$NN-SV, whose underlying classifier is equivalent to the standard weighted $k$NN classifier. We demonstrate that this weighted variant can be effectively reduced to the unweighted case by a duplication technique. Since naïve duplication could significantly increase the dataset size and computational burden, we further develop an efficient algorithm that exploits the structural properties of the weighted $k$NN models, thereby eliminating the additional computational costs associated with data duplication. In other words, the data duplication has been made *conceptual* and does not need to be materialized. Our algorithm runs in near-linear time $\mathcal{O}(n \log n)$, while the state-of-the-art methods either require a pseudo-polynomial time complexity of $\mathcal{O}(Wk^2n^2)$ [10] or settle for approximation by sampling from $n!$ permutations [8], where $n$ is the dataset size and $W$ is the maximum weight. We analyze the theoretical properties of the proposed D$k$NN-SV scheme, and empirically validate its effectiveness and efficiency against other $k$NN-SV variants. For example, our algorithm performs better at the task of noisy label detection.

The rest of the paper is organized as follows. First, we review the preliminaries in Section 2 and the unweighted $k$NN-SV in Section 3. Next, we introduce the new weighted D$k$NN-SV in Section 4, and develop an efficient algorithm for it in Section 5. We discuss the related work in Section 6. Finally, we empirically evaluate the proposed method in Section 7. We conclude in Section 8. All proofs are deferred to Appendix A.

## 2 Preliminaries

In this section, we review the framework for data valuation based on Shapley values (SV) and establish our notation for applying this concept to the $k$-nearest neighbor ($k$NN) model.

### 2.1 Cooperative game theory and Shapley values

The concept of data valuation can be elegantly formalized through the lens of cooperative game theory [12]. In this framework, we consider a collection of players who form coalitions to generate collective utility. Formally, a cooperative game consists of a pair $(I, \text{util})$, where $I = \{1, \ldots, n\}$ represents a set of $n$ players and $\text{util} : 2^I \to \mathbb{R}$ is a utility function that assigns a real value to each possible coalition $S \subseteq I$.

A central question in cooperative game theory concerns fair allocation: how should the total utility be distributed among individual players based on their contributions? The Shapley value, introduced by Lloyd Shapley [6], provides a rigorous solution to this problem. The Shapley value $s(i)$ for player $i$ represents their expected marginal contribution when joining the coalition in a random order. Let $\Pi$ denote the set of all permutations of players in $I$. For a permutation $\pi \in \Pi$, let $\pi_i$ represent the set of players that precede player $i$ in $\pi$. Then, the Shapley value equals

$$s(i) = \frac{1}{n!} \sum_{\pi \in \Pi} \left[ \text{util}(\pi_i \cup \{i\}) - \text{util}(\pi_i) \right]. \tag{1}$$

An equivalent expression represents the Shapley value as the average marginal contribution across all possible coalition formations, that is,

$$s(i) = \frac{1}{n} \sum_{S \subseteq I \setminus \{i\}} \binom{n-1}{|S|}^{-1} \left[ \text{util}(S \cup \{i\}) - \text{util}(S) \right]. \tag{2}$$

It is well known that the Shapley value is the *unique* allocation mechanism that satisfies the following desirable properties: *efficiency*, *symmetry*, *null player*, and *additivity* [6]. These properties make the Shapley values a principled approach for data valuation, relying on an axiomatic framework.

To apply the Shapley-value framework to the data-valuation setting, we interpret individual data points as players in a coalition game, and a performance measure of a model as a utility function. The value of the utility function for a coalition of players corresponds to the performance measure of the model trained on the respective subset of data. This framework allows for quantifying the contribution of each training data point to the overall model performance.

## 2.2 $k$NN-based Shapley values

In this section we discuss how the Shapley-value framework described above can be used for data valuation with a $k$NN model [8]. We refer to this method as $k$NN-SV. The idea is first presented for a single test data point, and then is extended to multiple test data points.

Consider a dataset $D$ with $n$ training data points, where each point $z = (x, y) \in D$ comprises $x \in \mathbb{R}^d$ and $y \in \mathcal{Y}$, where $\mathcal{Y}$ is the label space. Consider also a single test data point $z_{\text{test}} = (x_{\text{test}}, y_{\text{test}})$. We want to compute the Shapley value $s(z \mid z_{\text{test}})$ of each training point $z \in D$ with respect to the test point $z_{\text{test}}$. For each point $z$ we consider a rational weight $w(z \mid z_{\text{test}}) \in \mathbb{Q}_+$, often determined by the distance between $x$ and $x_{\text{test}}$ (see discussion below). For a subset $S \subseteq D$ of the training data, the $k$NN utility of $S$ is defined as

$$\text{util}(S) = \begin{cases} \frac{\sum_{i=1}^{\min\{|S|,k\}} w(z_{\alpha_i(S)} \mid z_{\text{test}}) \mathbb{1}(y_{\alpha_i(S)} = y_{\text{test}})}{\sum_{i=1}^{\min\{|S|,k\}} w(z_{\alpha_i(S)} \mid z_{\text{test}})}, & \text{if } |S| > 0 \\ \frac{1}{C}, & \text{if } |S| = 0, \end{cases} \tag{3}$$

where $C = |\mathcal{Y}|$ is the number of classes, $\alpha_i(S)$ is the index of the $i$-th closest point of $S$ to $x_{\text{test}}$, and $i$ is the *rank* of $z_{\alpha_i(S)}$ in $S$.

For defining the weight $w(z \mid z_{\text{test}})$, in the case of unweighted $k$NN, we simply set $w(z \mid z_{\text{test}}) = 1$. More generally, by using the distance $\text{dist}(z, z') = \|x - x'\|$, we can define the weight $w(z \mid z_{\text{test}})$ as the Gaussian kernel $w(z \mid z_{\text{test}}) = \mathcal{K}(\text{dist}(z, z_{\text{test}})) = \exp(-\text{dist}(z, z_{\text{test}})^2 / 2\sigma^2)$, where $\sigma$ measures the *width* of the kernel. Clearly, any other distance metric can be used.

Having defined the utility function for subsets $S \subseteq D$, the Shapley value of a data point $z \in D$ can be computed using Eq. (2). Formally, the Shapley value of a training data point $z \in D$ concerning a single test data point $z_{\text{test}}$ is defined as

$$s(z \mid z_{\text{test}}) = \frac{1}{n} \sum_{S \subseteq D \setminus \{z\}} \binom{n-1}{|S|}^{-1} \left[ \text{util}(S \cup \{z\}) - \text{util}(S) \right]. \tag{4}$$

In practice, when given multiple test data points, the data Shapley value of a data point $z$ can be straightforwardly extended as the average over all test points. That is, when given a test dataset $D_{\text{test}}$ of $n_{\text{test}}$ data points, we have

$$s(z) = \frac{1}{n_{\text{test}}} \sum_{z_{\text{test}} \in D_{\text{test}}} s(z \mid z_{\text{test}}). \tag{5}$$

## 3  Exact computation for unweighted $k$NN-SV

In this section, we introduce the analytical solution to unweighted $k$NN-SV, introduced by Jia et al. [8] and a subsequent note [13]. This solution results in a dramatic improvement in the time complexity for exact computation, from $\mathcal{O}(2^n)$ to $\mathcal{O}(dn + n \log n)$ for a single test point, over the naïve approach that enumerates all possible subsets $S$ of the dataset $D$.

We fix a test point $z_{\text{test}}$ throughout this section. Given a subset $S \subseteq D$, recall that $\alpha_i(S)$ is the index of the $i$-th closest element of $S$ to $z_{\text{test}}$. When the context is clear, we write $z_{\alpha_i(D)}$ as $z_i$ and $w(z_{\alpha_i(D)} \mid z_{\text{test}})$ as $w_i$ for simplicity. Similarly, we denote by $s_i$ the Shapley value $s(z_i \mid z_{\text{test}})$ for data point $z_i$.

We restate the main result from Wang and Jia [13] for unweighted $k$NN-SV. For ease of exposition, we assume $n \geq 2$ and $n \geq k$ throughout the paper.

**Theorem 1** (Wang and Jia [13]). *Assume $n \geq 2$ and $n \geq k$. For the unweighted $k$NN classifier, the Shapley value of a data point $z_i$ can be computed as follows.*

$$s_n = \frac{1}{n}\left(\mathbb{1}_n - \sum_{i=1}^{n-1}\frac{\mathbb{1}_i}{n-1}\right)\left(\sum_{j=1}^{k-1}\frac{1}{j+1}\right) + \frac{\mathbb{1}_n - C^{-1}}{n}, \tag{6}$$

*where $\mathbb{1}_i = \mathbb{1}[y_i = y_{test}]$, and for $i < n$,*

$$s_i = s_{i+1} + \frac{\mathbb{1}_i - \mathbb{1}_{i+1}}{n-1}\left(\sum_{j=1}^{k}\frac{1}{j} + \frac{1}{k}\left(\frac{\min\{k,i\}(n-1)}{i} - k\right)\right). \tag{7}$$

Following Theorem 1, we can compute the Shapley values by first sorting the data points by increasing distance to $z_{\text{test}}$, and then iteratively computing the Shapley values in the reverse order, as specified by Eq. (6) and Eq. (7).

However, extending the previous result to the weighted $k$NN is challenging. Given a subset $S$, we refer to the quantity $\text{util}(S \cup \{z_i\}) - \text{util}(S)$ as the *marginal contribution* (MC) of $z_i$ to $S$. The key principle behind the result in Theorem 1 is that for unweighted $k$NN, the MC can only take a few distinct values, which turns $k$NN-SV problem into a combinatorial counting problem. For example, for any $|S| \geq k$, we have

$$\text{util}(S \cup \{z_i\}) = \text{util}(S) \quad \text{or} \quad \text{util}(S \cup \{z_i\}) - \text{util}(S) = \frac{1}{k}(\mathbb{1}[y_i = y_{\text{test}}] - \mathbb{1}[y_{\alpha_k(S)} = y_{\text{test}}]),$$

the latter of which only takes three possible distinct values. Thus, one can simply count the number of subsets $S$ for each distinct MC value, and aggregate the results, without evaluating the MC for all possible subsets. However, for weighted $k$NN, as in Eq. (3), the MC may take arbitrary values due to the weights and the normalization term.

## 4 Weighted $k$NN-SV via data duplication

In this section, we show how to overcome the challenge in the weighted $k$NN-SV problem. We do so by introducing a new variant and reducing it to the unweighted case using a novel data-duplication technique. We relate the new variant to the concept of *Owen values* [14], and prove adherence to axiomatic properties. We further compare it with other $k$NN-SV variants analytically in Section 4.2.

Our main idea is to create $w(z \mid z_{\text{test}}) - 1$ copies of every point $z \in D$. For ease of exposition, we assume that all weights are integers. This assumption can be easily removed by scaling the weights appropriately. Let the new dataset containing $D$ and its copies be $D'$. We will then apply the unweighted $k$NN classifier to $D'$ with parameter $k'$.

To obtain an equivalent weighted $k$NN classifier, we adaptively set $k'$ for each test point $z_{\text{test}}$ [15]. Specifically, we adjust $k'$ so that it includes the top-$k$ nearest neighbors of $z_{\text{test}}$ in $D$ and their copies. Formally, we set

$$k' = \sum_{i=1}^{k} w(z_{\alpha_i(D)} \mid z_{\text{test}}). \tag{8}$$

We show that this leads to an equivalent weighted $k$NN classifier.

**Proposition 2.** *Running unweighted $k$NN classifier on a duplicated dataset $D'$ with parameter $k'$ defined in Eq. (8) is equivalent to running weighted $k$NN classifier on the original dataset $D$ for any test point $z_{test}$.*

Fixing a test point $z_{\text{test}}$, we define the utility of unweighted $k$NN on $D'$ for any subset $S'$ of $D'$ as

$$\text{util}'(S') = \begin{cases} \frac{\sum_{i=1}^{\min\{|S'|,k'\}} \mathbb{1}(y_{\alpha_i(S')}=y_{\text{test}})}{\min\{|S'|,k'\}}, & \text{if } |S'| > 0 \\ \frac{1}{C}, & \text{if } |S'| = 0. \end{cases} \tag{9}$$

The Shapley value of a data point $z \in D'$ concerning a given test point $z_{\text{test}}$ is then simply

$$s'(z \mid z_{\text{test}}) = \frac{1}{n'} \sum_{S' \subseteq D' \setminus \{z\}} \binom{n'-1}{|S'|}^{-1} \left[ \text{util}'(S' \cup \{z\}) - \text{util}'(S') \right]. \tag{10}$$

The D$k$NN-SV of a data point $z \in D$ is the sum of the Shapley values of all its copies in $D'$, i.e.,

$$\phi(z \mid z_{\text{test}}) = \sum_{z' \in D': z'=z} s'(z' \mid z_{\text{test}}) = w(z \mid z_{\text{test}})s'(z \mid z_{\text{test}}). \tag{11}$$

Note that duplicating data points according to their weights may result in a dataset $D'$ much larger than $D$, which may cause significant scalability issues. We will address this challenge in Section 5 by proposing an efficient algorithm that avoids materializing $D'$.

## 4.1 Duplication-based weighted $k$NN-SV as group values

An alternative way to interpret the D$k$NN-SV $\phi(z \mid z_{\text{test}})$ is to view a data point $z$ and its copies as a "group" with $z$ being the representative of the group, and $\phi(z \mid z_{\text{test}})$ measuring the value of the whole group. This is similar to the classic concept of *Owen values* [14], which characterizes the value of players in a game with *coalition structure*.

Formally, to compute the Owen values, we are given non-overlapping groups $\mathcal{G} = \{G_1, \ldots, G_m\}$ such that $\cup_{i=1}^m G_i = D'$ and $G_i \cap G_j = \emptyset$ for all $i \neq j$. Owen values are calculated by considering permutations of players that are compatible with the group structure. A permutation $\pi$ of the player set $D'$ is called *group-compatible* if players within the same group $G_i$ appear consecutively in the permutation. The Owen value of a player $z'$ is its average marginal contribution over all group-compatible permutations, computed in a way similar to Eq. (1). The only difference is that the Owen value sums over all group-compatible permutations instead of all permutations. The Owen value of a group $G_i$ is then the sum of the Owen values of all players $z' \in G_i$.

In our setting, each group is a set of copies of a single point. The main difference is that we do not require all players in a group to always act as one unit, and thus do not require the permutations to be group-compatible. Instead, we allow each player (data point) to act independently in the game, and eventually aggregate their contributions to obtain the group value in the same manner. This relaxation allows us to design an efficient algorithm. Our approach can also be justified axiomatically. These axiomatic properties ensure that data valuation is conducted in a fair, theoretically sound, and interpretable manner for downstream tasks. Moreover, additional properties about symmetry between $z$ and its copies enable us to devise a fast algorithm without materializing $D'$; see Section 5.

**Theorem 3.** *The group value $\phi(z \mid z_{test})$ defined in Eq. (11) satisfies the following axioms:*

1. **Efficiency:** $\sum_{z \in D} \phi(z \mid z_{test}) = util'(D') - util'(\emptyset) = util(D) - util(\emptyset)$, *that is, the sum of all group values equals the total utility.*

2. **Symmetry:** *If $z_1$ and $z_2$ are such that $util'(S' \cup \{z_1'\}) = util'(S' \cup \{z_2'\})$ for all $S' \subseteq D' \setminus \{z_1', z_2'\}$ where $z_1'$ and $z_2'$ are copies of $z_1$ and $z_2$ respectively, then $\phi(z_1 \mid z_{test})/w(z_1 \mid z_{test}) = \phi(z_2 \mid z_{test})/w(z_2 \mid z_{test})$.*

3. **Dummy Player:** *If $z$ is such that $util'(S' \cup \{z'\}) = util'(S')$ for all $S' \subseteq D' \setminus \{z'\}$ where $z'$ is a copy of $z$, then $\phi(z \mid z_{test}) = 0$.*

4. **Additivity:** *If $util_1'$ and $util_2'$ are two utility functions, and $\phi_1$ and $\phi_2$ are their corresponding values, then the value $\phi$ corresponding to $util_1' + util_2'$ satisfies $\phi = \phi_1 + \phi_2$.*

Theorem 3 can be easily extended to more general settings, where each group is formed by an arbitrary subset of players instead of a set of copies.

## 4.2 Comparison with other $k$NN-SV

**Comparison with standard weighted $k$NN-SV.** Although Proposition 2 shows that the duplication strategy provides an equivalent weighted $k$NN classifier, it may not produce the same Shapley values. This is because we need equivalent classification over all possible subsets instead of merely the entire dataset $D$. More generally, we show that their values remain different regardless of the value of parameter $k'$. We give a counterexample below. The intuition is that the effect of increasing the weight of a point and increasing its number of copies on its value is similar over most subsets, but may differ over certain subsets.

**Proposition 4.** *In the genral case, the group Shapley values in Eq.* (11) *may be different from the Shapley values of weighted $k$NN-SV in Eq.* (4).

Hence, our duplication strategy offers a new variant for computing data Shapley values for weighted $k$NN-SV. Due to the lack of a closed-form derivation for the vanilla weighted $k$NN-SV, we compare the different methods empirically in the experiments, and show that they are correlated, however, the values in the proposed approach can be computed more efficiently.

**Comparison with the scaled unweighted $k$NN-SV.** One might also wonder how the group values $\phi(z \mid z_{\text{test}})$ compare to simply scaling the unweighted $k$NN-SV values $s(z \mid z_{\text{test}})$ in the original dataset $D$. A natural scaling approach might be $s(z \mid z_{\text{test}}) \cdot w(z \mid z_{\text{test}})$. However, such scaling is fundamentally different from our duplication strategy, and fails to capture the inherent weighted nature of the weighted $k$NN-SV, as illustrated in the analysis below.

Given an arbitrary test point $z_{\text{test}}$, we first examine the scaled value $\tilde{s}_n$ of the farthest point $z_n$.

$$\tilde{s}_n = s_n w_n \frac{n}{n'} = \frac{w_n}{n'}\left(\mathbb{1}_n - \sum_{i=1}^{n-1}\frac{\mathbb{1}_i}{n-1}\right)\left(\sum_{j=1}^{k-1}\frac{1}{j+1}\right) + w_n \frac{\mathbb{1}_n - \frac{1}{C}}{n'},$$

where $w_n = w(z_n \mid z_{\text{test}})$ and $\mathbb{1}_i = \mathbb{1}(y_i = y_{\text{test}})$. To simplify the analysis, we set $k'$ to be the same as $k$. If we compare the scaled value $\tilde{s}_n$ with the group value $\phi(z_n \mid z_{\text{test}})$, we have

$$\tilde{s}_n - \phi(z_n \mid z_{\text{test}}) = \frac{w_n}{n'}\left(\left[\mathbb{1}_n - \underbrace{\sum_{i=1}^{n-1}\frac{\mathbb{1}_i}{n-1}}_{\text{unweighted avg}}\right] - \left[\mathbb{1}_n - \underbrace{\frac{(w_n - 1)\mathbb{1}_n + \sum_{i=1}^{n-1}w_i\mathbb{1}_i}{n'-1}}_{\text{weighted avg}}\right]\right)\left(\sum_{j=1}^{k-1}\frac{1}{j+1}\right).$$

Thus, the difference is driven by the choice of the reference mean used in the comparison. Define unweighted and weighted reference means as

$$\mu_{\text{unw}} := \sum_{i=1}^{n-1}\frac{\mathbb{1}_i}{n-1}, \qquad \text{and} \qquad \mu_{\text{w}} := \frac{(w_n - 1)\mathbb{1}_n + \sum_{i=1}^{n-1}w_i\mathbb{1}_i}{n'-1}.$$

Then the scaled value $\tilde{s}_n$ uses the deviation $\mathbb{1}_n - \mu_{\text{unw}}$, which measures the contribution of $z_n$ by the deviation of its label from this unweighted reference mean, whereas the group value considers the deviation from the weighted reference mean by using $\mathbb{1}_n - \mu_{\text{w}}$.

Moving on to the scaled value of the $i$-th closest point $z_i$, we have

$$\tilde{s}_i := s_i w_i \frac{n}{n'}$$

$$= s_{i+1}w_i\frac{n}{n'} + w_i\frac{n}{n-1}\frac{\mathbb{1}_i - \mathbb{1}_{i+1}}{n'}\left(\sum_{j=1}^{k}\frac{1}{j} + \frac{1}{k}\left(\frac{\min\{k,i\}(n-1)}{i} - k\right)\right)$$

$$\approx s_{i+1}w_i\frac{n}{n'} + w_i\frac{\mathbb{1}_i - \mathbb{1}_{i+1}}{n'-1}\left(\sum_{j=1}^{k}\frac{1}{j} + \frac{1}{k}\left(\frac{\min\{k,i\}(n-1)}{i} - k\right)\right),$$

where the last approximation follows by taking $\frac{n-1}{n}\frac{n'}{n'-1} \approx 1$. Then, if we compare the scaled value $\tilde{s}_i$ with the group value $\phi(z_i \mid z_{\text{test}})$, we have

$$\tilde{s}_i - \phi(z_i \mid z_{\text{test}})$$

---

**Algorithm 1:** Fast algorithm for duplicate-based weighted $k$NN-SV

---
    **Input:** Integer $k$, weight function $w$, datasets $D$ and $D_{test}$

1  Initialize $\phi_z$ with default value 0 for every $z \in D$

2  **for** $z_{test} \in D_{test}$ **do**

3     Let $z_1, \ldots, z_n$ be the points in $D$ sorted by increasing distance to $z_{test}$

4     $n' \leftarrow \sum_{i=1}^{n} w(z_i \mid z_{test})$

5     $k' = \sum_{i=1}^{k} w(z_i \mid z_{test})$

6     $s'_{z_n} \leftarrow \frac{1}{n'}\left(\mathbb{1}_n - \frac{(w_n-1)\mathbb{1}_n + \sum_{i=1}^{n-1} w_i \mathbb{1}_i}{n'-1}\right)\left(\sum_{j=1}^{k'-1} \frac{1}{j+1}\right) + \frac{\mathbb{1}_n - 1/C}{n'}$, where

        $w_i = w(z_i \mid z_{test})$ and $\mathbb{1}_i = \mathbb{1}[y_i = y_{test}]$

7     **for** $i = n-1, \ldots, 1$ **do**

8         $i' \leftarrow \sum_{j=1}^{i} w(z_j \mid z_{test})$

9         $s'_{z_i} \leftarrow s'_{z_{i+1}} + \frac{\mathbb{1}_i - \mathbb{1}_{i+1}}{n'-1}\left(\sum_{j=1}^{k'} \frac{1}{j} + \frac{1}{k'}\left(\frac{\min(k',i')(n'-1)}{i'} - k'\right)\right)$

10    **for** $i = n, \ldots, 1$ **do**

11       $\phi_{z_i} \leftarrow \phi_{z_i} + w(z_i \mid z_{test}) s'_{z_i}$

12 **for** $z \in D$ **do**

13    $\phi_z \leftarrow \phi_z / n_{test}$

14 Return values $\{\phi_z\}_{z \in D}$

---

$$\approx w_i\left(\frac{s_{i+1}n}{n'} - \frac{\phi(z_{i+1})}{w_{i+1}}\right) + w_i\frac{\mathbb{1}_i - \mathbb{1}_{i+1}}{n'-1}\frac{1}{k}\left(\frac{\min\{k,i\}(n-1)}{i} - \frac{\min\{k,i'\}(n'-1)}{i'}\right).$$

First, the difference between these two values accumulates from the previous $(i+1)$-th point. In addition, the contribution from the $i$-th point to the scaled value $\tilde{s}_i$ is roughly weighted by $\frac{w_i(n-1)}{i(n'-1)}$, while the contribution to the group value $\phi(z_i \mid z_{\text{test}})$ is roughly weighted by $\frac{w_i}{i'}$. That is, the former scales up every rank $i$ uniformly by a factor of $\frac{n'-1}{n-1}$, while the latter uses the rank $i'$ of the $i$-th point in the duplicated dataset $D'$. Therefore, the scaled value tends to amplify the contribution of nearest neighbors with a large weight more than the group value.

## 5   Fast algorithm for duplication-based weighted $k$NN-SV

Recall that we duplicate data points in $D$ according to their weights to form a new dataset $D'$. However, the size of $D'$ may be much larger than that of $D$. This poses a significant challenge to the computational cost of the recursive computation of the Shapley values. Specifically, it requires $\mathcal{O}(n' \log n')$ time for each given test point, where $n'$ is the size of $D'$. In this section, we propose a fast algorithm for the duplication-based weighted $k$NN-SV, with a time complexity as small as $\mathcal{O}(n \log n)$. The algorithm successfully obtains the Shapley values while avoiding materializing the duplicated dataset $D'$.

The proposed algorithm is displayed in Algorithm 1. The key idea is to leverage the *symmetry* property of the Shapley values to avoid materializing the duplicated data points.

**Lemma 5.** *Fix a test point $z_{test}$. Let $z$ be a point in $D$. Then, every copy $z'$ of $z$ has the same Shapley value $s'(z' \mid z_{test})$ as that of $z$. This continues to hold when the values are obtained by applying the recursive formula in Theorem 1 with an arbitrary order among $z$ and its copies.*

Therefore, when applying the recursive formula in Theorem 1, we only process every point $z$ in $D$ once, and skip its copies by directly multiplying the Shapley value by the number of its copies. This is valid as a result of Lemma 5; imagine we are taking an order where $z$ is behind all its copies and processed first.

Note that the partial sum of the harmonic series can be accurately approximated by the Euler-Maclaurin formula [16], or retrieved from a pre-computed table. Asymptotically, the Euler-Maclaurin

formula [16] $F(j)$ below converges to the harmonic series $H_j = \sum_{i=1}^{j} \frac{1}{i}$ as $j \to \infty$.

$$F(j) = \ln(j) + \gamma + \frac{1}{2j} - \frac{1}{12j^2},$$

where $\gamma \approx 0.5772156649$ is the Euler-Mascheroni constant. In practice, we adopt the following more accurate approximation scheme. When $j$ is a small constant (e.g., $j \leq 1000$), we can afford to compute $H_j$ exactly. Otherwise, we use $F(j) - F(1000) + H_{1000}$ as an approximation (in $O(1)$ time where $H_{1000}$ is pre-computed) to eliminate the noticeable error when $j$ is relatively small. We verify that the absolute error is at most $4.167 \times 10^{-8}$ for $j$ up to 1 million and continues to decrease as $j$ increases. We believe this approximation is sufficient for our purpose.

Hence, the total time complexity is dominated by the sorting step, which is $\mathcal{O}(n \log n)$. In addition, distance computations cost $\mathcal{O}(dn)$ time. We summarize the results in the following theorem.

**Theorem 6.** *Algorithm 1 computes duplication-based weighted $k$NN-SV in $\mathcal{O}(dn + n \log n)$ time.*

## 6 Related work

**Data Valuation.** Data valuation aims to assign importance scores to training examples [3, 4, 5]. The dominant approaches are based on the concept of leave-one-out (LOO), which measures the marginal contribution of a data point to the utility function (e.g., model accuracy) when it is removed from the training procedure. DataShapley [1] and its variants such as BetaShapley [17], DataBanzhaf [18], least core [19], are all based on the LOO principle, but differ in the way the marginal contributions are aggregated. We discuss several notable options beyond Shapley values below. Feldman and Zhang [20] simulate the data values by LOO retraining albeit constrained on a small sample of the training data, while DataModels [21] sacrifice the exactness of LOO to achieve better scalability by model predictions. Another line of popular methods is gradient-based. TracIn [22] estimates the importance of a training example by tracing the change in the test loss caused by the example during the training process. Variations of influence functions [23, 24] have their roots in robust statistics [25], and offer a gradient-based approximation of the LOO values.

**Shapley Values.** Shapley values [6] originated in cooperative game theory as a method for fairly distributing gains among players, and have been widely adopted in multiple fields such as economics [26]. Computing exact Shapley values is well-known to be expensive, i.e., #**P**-hard in certain games [7]. This computational challenge has motivated various approximation techniques, including mostly Monte Carlo sampling [27, 28, 29, 30], and specialized algorithms for specific games [31]. Our work falls into the latter category. Early work on Shapley values with exogenous coalitions distributes among players from a group $G$ the utility util$(G)$, respecting the so-called *relative efficiency* axiom. Owen [14] and subsequent work [32] further consider a game between coalitions. Our duplication technique creates natural coalitions in the game.

**$k$NN-based Shapley Values ($k$NN-SV).** The $k$-Nearest Neighbor ($k$NN) model provides a unique opportunity for efficient computation of data Shapley values. Jia et al. [8] are the first to discover an efficient algorithm in time $\mathcal{O}(n_{test} n \log n)$ for unweighted $k$NN-SV. Wang and Jia [13] provide refinements to the unweighted $k$NN utility function. The weighted $k$NN case turns out to be more challenging due to the normalization factor in the utility function. Wang et al. [10] propose a time-consuming dynamic-programming algorithm for weighted $k$NN-SV with a hard-label utility function. Our work offers a more efficient approach for weighted $k$NN-SV.

## 7 Experiments

We investigate the following research questions in the experiments: (1) How does Algorithm 1 perform compared with the existing methods? We study this question with a task of noisy label detection. See Section 7.1. (2) How does the D$k$NN-SV deviate from those of the unweighted and weighted $k$NN-SV formulations? We visualize and compare them in Section 7.2. (3) How is the scalability of Algorithm 1? We study this in Fig. 1b and further in Appendix B.3. (4) What is the effect of the parameters on Algorithm 1? We study this in Fig. 1c and more in Appendix B.4.

*Datasets.* We evaluate the proposed methods on 11 datasets, whose statistics are listed in Table A1. The size of the datasets ranges from 5K to 1M. Many of them are chosen to be of a moderate size, so

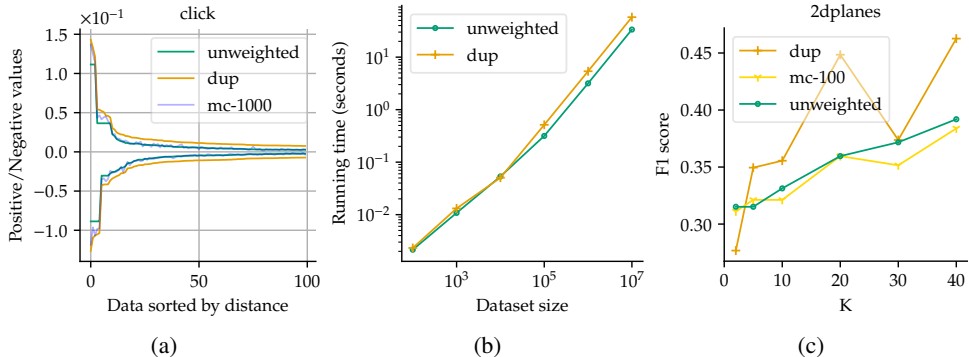

(a)                      (b)                      (c)

Figure 1: Visualization of the Shapley values of different methods concerning a random test point (Fig. 1a). The running time on a random 1-dimensional dataset of different sizes (Fig. 1b). The effect of the parameter $k$ on the performance of different methods (Fig. 1c).

as to allow us to compare with more costly baselines. By default, we randomly select 1% the data up to 100 points as the testing set.

*Baselines.* We include the following baselines in the experiments: (1) unweighted $k$NN-SV (*unweighted*), (2) scaled unweighted $k$NN-SV (*scaled*) in Section 4.2, (3) weighted $k$NN-SV (*mc*) by fast Monte Carlo sampling [8] with different number of samples, (4) random selection (*random*), and finally (5) our duplication-based D$k$NN-SV in Algorithm 1 (*dup*). The hard-label weighted $k$NN-SV [10] is not included as it fails to finish within 5 hours on small datasets (with default discretization bits $n_{bits} = 3$), so we report comparison results with it on tiny datasets in Appendix B.1. Our code can be found at a Github repository.[2] By default, we run each algorithm three times and report the average results.

### 7.1 Noisy label detection

To evaluate the performance of different methods, we follow the setup in previous works [8, 13, 10] and use the task of noisy label detection. For each dataset, we randomly flip the labels of 5% of the training data points, which forms a noisy subset of size $n/20$. We predict the noisy subset by the top-$t$ data points with the lowest Shapley values. We set $t = 500$ for all methods. Intuitively, stronger data valuation methods should be able to detect noisy data points more accurately.

We tune the key parameter, kernel width $\sigma$, of all algorithms that use a kernel function as follows. We randomly select 5% of the training data as a validation set. We train a weighted $k$NN classifier on the rest of the data, and evaluate its accuracy on the validation set. We select the value $\sigma$ that gives the highest accuracy from a list of candidates. Note that no information about the noisy labels is leaked to the validation process.

The results are shown in Tables 1 and 2 with standard deviation. Our proposed algorithm *dup* consistently outperforms all the other methods on most datasets. Its running time is slightly higher than that of the unweighted $k$NN-SV due to the additional cost in weighting the data points, but still very efficient. The weighted $k$NN-SV (*mc*) achieves better performance as the number of samples increases; however, it only reaches the performance of unweighted $k$NN-SV when the sample number is as large as 1000 and is nearly 1000 times slower. The scaled $k$NN-SV (*scaled*) is comparable to the unweighted $k$NN-SV. All methods perform significantly better than random selection.

### 7.2 Visualization of Shapley values

We visualize the Shapley values of different methods with respect to a random test point $z_{\text{test}}$ as follows. We sort all data points by an increasing distance to $z_{\text{test}}$ and plot their Shapley values. Besides, we plot the positive and negative values separately. We limit the number of points to the top 100 as the rest are all close to zero.

---

[2]`https://github.com/Guangyi-Zhang/weighted-knnsv-via-duplication`

Table 1: F1 scores of different methods on the noisy label detection task. The best one is highlighted in bold, and the second best is underlined. '-' indicates a timeout.

| | unweighted | dup | scaled | random | mc-10 | mc-100 | mc-1000 |
|---|---|---|---|---|---|---|---|
| phoneme | **0.318** ± 0.074 | 0.283 ± 0.167 | 0.232 ± 0.090 | 0.032 | 0.225 | 0.298 | 0.318 ± 0.074 |
| wind | 0.286 ± 0.026 | **0.357** ± 0.017 | 0.277 ± 0.009 | 0.043 | 0.240 | 0.283 | 0.289 ± 0.017 |
| cpu | 0.507 ± 0.089 | **0.565** ± 0.014 | 0.444 ± 0.000 | 0.049 | 0.365 | 0.486 | 0.507 ± 0.096 |
| 2dplanes | 0.342 ± 0.021 | **0.368** ± 0.017 | 0.330 ± 0.027 | 0.055 | 0.226 | 0.339 | 0.351 ± 0.016 |
| apsfail | 0.746 ± 0.004 | **0.795** ± 0.007 | 0.737 ± 0.020 | 0.055 | 0.520 | 0.732 | 0.749 ± 0.003 |
| click | 0.070 ± 0.021 | **0.072** ± 0.033 | 0.059 ± 0.011 | 0.055 | 0.054 | 0.067 | 0.069 ± 0.017 |
| creditcard | 0.119 ± 0.040 | **0.132** ± 0.050 | 0.120 ± 0.039 | 0.055 | 0.102 | 0.118 | 0.118 ± 0.036 |
| fraud | 0.879 ± 0.003 | **0.888** ± 0.010 | 0.874 ± 0.019 | 0.055 | 0.737 | 0.853 | 0.881 ± 0.011 |
| pol | 0.420 ± 0.003 | **0.447** ± 0.021 | 0.425 ± 0.024 | 0.055 | 0.266 | 0.390 | 0.430 ± 0.000 |
| vehicle | **0.140** ± 0.007 | 0.126 ± 0.024 | 0.119 ± 0.031 | 0.055 | 0.111 | 0.128 | 0.138 ± 0.007 |
| poker | 0.150 ± 0.002 | **0.305** ± 0.001 | 0.137 ± 0.001 | 0.050 | 0.069 | - | - |

Table 2: Running time (seconds) of different methods on the noisy label detection task. The best one is highlighted in bold, and the second best is underlined. '-' indicates a timeout.

| | unweighted | dup | scaled | random | mc-10 | mc-100 | mc-1000 |
|---|---|---|---|---|---|---|---|
| phoneme | 1.2 ± 0.2 | 1.6 ± 0.2 | 1.3 ± 0.1 | **0.0** | 9.8 | 92.1 | 874.1 ± 25.5 |
| wind | 1.5 ± 0.0 | 2.4 ± 0.2 | 1.9 ± 0.1 | **0.0** | 14.9 | 129.2 | 1278.2 ± 26.6 |
| cpu | 2.3 ± 0.2 | 3.7 ± 0.2 | 3.2 ± 0.3 | **0.0** | 22.3 | 206.7 | 1975.4 ± 12.3 |
| 2dplanes | 3.5 ± 0.3 | 5.8 ± 0.0 | 4.7 ± 0.2 | **0.0** | 33.5 | 307.5 | 2961.1 ± 69.8 |
| apsfail | 3.6 ± 0.4 | 5.7 ± 0.6 | 4.7 ± 0.5 | **0.0** | 33.7 | 299.0 | 2940.1 ± 47.7 |
| click | 3.4 ± 0.3 | 5.5 ± 0.6 | 4.5 ± 0.3 | **0.0** | 33.9 | 302.4 | 2931.1 ± 39.7 |
| creditcard | 3.4 ± 0.4 | 5.4 ± 0.4 | 4.7 ± 0.4 | **0.0** | 33.9 | 305.7 | 2952.4 ± 49.5 |
| fraud | 3.4 ± 0.2 | 5.3 ± 0.3 | 4.5 ± 0.3 | **0.0** | 33.9 | 300.0 | 2923.2 ± 27.9 |
| pol | 3.6 ± 0.3 | 5.6 ± 0.9 | 4.7 ± 0.5 | **0.0** | 33.1 | 297.2 | 2937.4 ± 29.1 |
| vehicle | 3.5 ± 0.0 | 5.6 ± 0.8 | 4.7 ± 0.3 | **0.0** | 34.6 | 307.2 | 2948.2 ± 4.2 |
| poker | 368.7 ± 41.9 | 637.4 ± 74.3 | 496.5 ± 36.7 | **0.0** | 3606.3 | - | - |

The representative visualization is shown in Fig. 1a and more in Appendix B.2. All methods follow a similar pattern and assign a larger absolute values to nearer points. One common property that is shared by *dup* and *mc* is that they are able to differentiate the nearest points, while the unweighted $k$NN-SV often assigns an equal value to, for example, the top 10 points.

## 8 Conclusion

In this paper, we introduced a novel variant of weighted $k$NN-SV that leverages a duplication technique to reduce the weighted case to the unweighted one effectively. This variant successfully captures the weighted nature of the $k$NN models and maintains the desirable axiomatic properties of Shapley values, while being amenable to efficient computation.

We discuss limitations and directions for future work. Our method does not solve the original weighted $k$NN-SV problem but rather a new variant of it. Though the resulting data values can be justified axiomatically, they are not Shapley values. Several promising directions for future work include extending our approach to $k$NN regression tasks, and investigating the applicability of our duplication technique to other $k$NN-related scenarios.

## Acknowledgments and Disclosure of Funding

This research is supported by the ERC Advanced Grant REBOUND (834862), the Swedish Research Council project ExCLUS (2024-05603), the Wallenberg AI, Autonomous Systems and Software Program (WASP) funded by the Knut and Alice Wallenberg Foundation, Guangdong Provincial College Youth Innovative Talent Project (Grant No. 2025KQNCX075), Natural Science Foundation of Top Talent of SZTU (Grant No. GDRC202520), SZTU University Research Project (No. 20251061020002), and fundamental research funds for the central universities of Ministry of Education of China (SWU-KR24043).

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

# A Missing proofs

**Theorem 3.** *The group value $\phi(z \mid z_{test})$ defined in Eq. (11) satisfies the following axioms:*

1. **Efficiency:** $\sum_{z \in D} \phi(z \mid z_{test}) = util'(D') - util'(\emptyset) = util(D) - util(\emptyset)$, *that is, the sum of all group values equals the total utility.*

2. **Symmetry:** *If $z_1$ and $z_2$ are such that $util'(S' \cup \{z_1'\}) = util'(S' \cup \{z_2'\})$ for all $S' \subseteq D' \setminus \{z_1', z_2'\}$ where $z_1'$ and $z_2'$ are copies of $z_1$ and $z_2$ respectively, then $\phi(z_1 \mid z_{test})/w(z_1 \mid z_{test}) = \phi(z_2 \mid z_{test})/w(z_2 \mid z_{test})$.*

3. **Dummy Player:** *If $z$ is such that $util'(S' \cup \{z'\}) = util'(S')$ for all $S' \subseteq D' \setminus \{z'\}$ where $z'$ is a copy of $z$, then $\phi(z \mid z_{test}) = 0$.*

4. **Additivity:** *If $util_1'$ and $util_2'$ are two utility functions, and $\phi_1$ and $\phi_2$ are their corresponding values, then the value $\phi$ corresponding to $util_1' + util_2'$ satisfies $\phi = \phi_1 + \phi_2$.*

*Proof of Theorem 3.* We prove each property separately.

**Efficiency:** It directly follows from the efficiency of Shapley values $s'(z' \mid z_{test})$ in the duplicated game. The equality $util'(D') = util(D)$ holds because of Proposition 2.

$$\sum_{z \in D} \phi(z \mid z_{test}) = \sum_{z \in D} \sum_{z' \in D' : z' = z} s'(z' \mid z_{test}) = \sum_{z' \in D'} s'(z' \mid z_{test}) = util'(D') - util'(\emptyset).$$

**Symmetry:** If two points $z_1$ and $z_2$ make identical marginal contributions through their copies, then by the symmetry of Shapley values, each copy receives the same value, i.e., $s'(z_1' \mid z_{test}) = s'(z_2' \mid z_{test})$. Thus,

$$\frac{\phi(z_1 \mid z_{test})}{w(z_1 \mid z_{test})} = \frac{w(z_1 \mid z_{test}) s'(z_1' \mid z_{test})}{w(z_1 \mid z_{test})} = s'(z_1' \mid z_{test}) = s'(z_2' \mid z_{test}) = \frac{\phi(z_2 \mid z_{test})}{w(z_2 \mid z_{test})}.$$

**Dummy Player:** If a point $z$ contributes nothing through any of its copies, then by the dummy player property of Shapley values, $s'(z' \mid z_{test}) = 0$ for all copies $z'$ of $z$. Therefore,

$$\phi(z \mid z_{test}) = w(z \mid z_{test}) s'(z \mid z_{test}) = w(z \mid z_{test}) \cdot 0 = 0.$$

**Additivity:** Let $util_1'$ and $util_2'$ be two utility functions with corresponding values $\phi_1$ and $\phi_2$. Let $s_1'$ and $s_2'$ be the Shapley values in the duplicated games for $util_1'$ and $util_2'$. By the additivity of Shapley values, $s'(z') = s_1'(z') + s_2'(z')$ for the combined utility $util' = util_1' + util_2'$. Therefore:

$$\begin{aligned}
\phi(z \mid z_{test}) &= w(z \mid z_{test}) s'(z \mid z_{test}) \\
&= w(z \mid z_{test})(s_1'(z \mid z_{test}) + s_2'(z \mid z_{test})) \\
&= w(z \mid z_{test}) s_1'(z \mid z_{test}) + w(z \mid z_{test}) s_2'(z \mid z_{test}) \\
&= \phi_1(z \mid z_{test}) + \phi_2(z \mid z_{test}).
\end{aligned}$$

$\square$

**Proposition 4.** *In the genral case, the group Shapley values in Eq. (11) may be different from the Shapley values of weighted $k$NN-SV in Eq. (4).*

*Proof of Proposition 4.* Consider a dataset $D = \{z_1, \ldots, z_n\}$ where all points share $z_{test}$'s label. Note that since all points have the correct label, the marginal contribution of adding any point to a subset $S$ is non-zero only when $S$ is empty, regardless of the value of $k$. Let $w(z_1) = 2$ and $w(z_i) = 1$ for $i \neq 1$.

We first discuss the weighted $k$NN-SV on $D$. It is easy to see that

$$s_1 = \frac{1}{n}\left(1 - \frac{1}{C}\right).$$

Next, we consider the unweighted $k$NN-SV on $D'$. By symmetry, we have $s_{z_1}' = s_{z_1'}'$, so

$$\phi(z_1) = s_{z_1}' + s_{z_1'}' = \frac{2}{n+1}\left(1 - \frac{1}{C}\right).$$

Equality $s_1 = \phi(z_1)$ holds *only* when $n = 1$. For $n > 1$ it holds $\frac{2}{n+1} > \frac{1}{n}$.

$\square$

Table A1: Statistics of the datasets used in the experiments.

| Dataset | $n$ | $d$ | $|\mathcal{Y}|$ |
|---|---|---|---|
| phoneme | 5404 | 5 | 2 |
| wind | 6574 | 14 | 2 |
| cpu | 8192 | 21 | 2 |
| 2dplanes | 10000 | 10 | 2 |
| apsfail | 10000 | 170 | 2 |
| click | 10000 | 11 | 2 |
| creditcard | 10000 | 23 | 2 |
| fraud | 10000 | 30 | 2 |
| pol | 10000 | 48 | 2 |
| vehicle | 10000 | 100 | 2 |
| poker | 1000000 | 10 | 10 |

**Lemma 5.** *Fix a test point $z_{test}$. Let $z$ be a point in $D$. Then, every copy $z'$ of $z$ has the same Shapley value $s'(z' \mid z_{test})$ as that of $z$. This continues to hold when the values are obtained by applying the recursive formula in Theorem 1 with an arbitrary order among $z$ and its copies.*

*Proof of Lemma 5.* The first statement is a direct consequence of the symmetry property of the Shapley values. The second statement follows by observing that any two copies of $z$ share the same label, which turns the second term in Eq. (7) into zero. Thus, their values do not depend on the order of $z$ and its copies. $\qquad\square$

**Theorem 6.** *Algorithm 1 computes duplication-based weighted $k$NN-SV in $\mathcal{O}(dn + n \log n)$ time.*

*Proof of Theorem 6.* Following Lemma 5, we take an order where each point $z$ is behind all its copies. Then we apply the recursive formula in Theorem 1 to compute the Shapley values. After processing each point $z \in D$, we can skip its copies by multiplying $s'(z \mid z_{test})$ by the number of its copies, as instructed by Lemma 5. Thus, no copies need to be materialized, and the time complexity is dominated by the sorting step, which is $\mathcal{O}(dn + n \log n)$. $\qquad\square$

# B    Additional experimental details

**Experimental Environment.** All algorithms were implemented in Python 3.11. All experiments were carried out on a Linux server equipped with 64 CPUs of Intel(R) Xeon(R) Platinum 8358P CPU @ 2.60 GHz and 1511 GB RAM.

See Table A1 for the statistics of the datasets used in the experiments.

## B.1    Comparison with hard-label weighted $k$NN-SV

Instead of resorting to the approximate version, we choose to compare with the exact version of this baseline [10] over small datasets. The hard-label weighted $k$NN-SV is denoted as *dp* below. We set the data size to be 300 by randomly sampling from the original datasets. We tune the parameter of kernel width in the same fashion as in the main experiments. Very surprisingly, as reported in Table A2, the *dp* is only slightly better than random guessing for the task of noisy label detection, and is far behind the performance of the unweighted $k$NN-Shapley and our method. To make sure this is not caused by bugs in the code, we further verify that its values are indeed consistent with those by Monte-Carlo sampling with a hard-label utility function.

After careful inspection, we identify the key reason: a hard-label utility function may be less suitable for the task of noisy label detection, or require stronger signal from distance-based weights. Unlike the soft-label utility, it is unable to capture the fine-grained contribution of a data point. It requires the noisy point to be a game changer for the prediction of its neighbors in order to be considered harmful, which is not the case for most mislabeled points. A mislabeled point is often surrounded by well-labeled points, and its ability to change the prediction of its neighbors is often negligible. Note

Table A2: F1 scores of different methods on the noisy label detection task. The best one is highlighted in bold, and the second best is underlined.

| | dup | unweighted | random | dp |
|---|---|---|---|---|
| 2dplanes | $\underline{0.232} \pm 0.090$ | $\mathbf{0.250} \pm 0.092$ | 0.036 | $0.071 \pm 0.058$ |
| apsfail | $\mathbf{0.786} \pm 0.143$ | $\underline{0.696} \pm 0.090$ | 0.036 | $0.071 \pm 0.058$ |
| click | $\mathbf{0.036} \pm 0.041$ | $\underline{0.036} \pm 0.041$ | 0.036 | $0.036 \pm 0.041$ |
| cpu | $\mathbf{0.607} \pm 0.071$ | $\underline{0.536} \pm 0.149$ | 0.036 | $0.071 \pm 0.101$ |
| creditcard | $\underline{0.107} \pm 0.124$ | $\mathbf{0.125} \pm 0.036$ | 0.036 | $0.054 \pm 0.068$ |
| fraud | $\underline{0.821} \pm 0.137$ | $\mathbf{0.893} \pm 0.071$ | 0.036 | $0.036 \pm 0.041$ |
| phoneme | $\mathbf{0.321} \pm 0.092$ | $\underline{0.321} \pm 0.041$ | 0.000 | $0.018 \pm 0.036$ |
| pol | $\mathbf{0.339} \pm 0.236$ | $\underline{0.339} \pm 0.147$ | 0.036 | $0.071 \pm 0.058$ |
| vehicle | $\underline{0.143} \pm 0.154$ | $\mathbf{0.179} \pm 0.189$ | 0.036 | $0.107 \pm 0.092$ |
| wind | $\mathbf{0.250} \pm 0.092$ | $\underline{0.214} \pm 0.154$ | 0.089 | $0.036 \pm 0.041$ |
| poker | $\mathbf{0.071} \pm 0.000$ | $\underline{0.071} \pm 0.000$ | 0.054 | $0.000 \pm 0.000$ |

that resorting to a hard-label utility function is the key modification that enables the DP algorithm in Wang et al. [10] to work.

## B.2 Visualization of the Shapley values

See Fig. A1 for more visualization of the Shapley values.

## B.3 Scalability

We compare the running time of *unweighted* and *dup* with different dataset sizes. We vary the size of a random 1-dimensional dataset $D$ from 100 to 10 M. The results are shown in Fig. 1b. As we can see, the running time of *dup* follows that of the unweighted $k$NN-SV closely, confirming its near-linear time complexity.

## B.4 Effect of the parameter $k$

We examine the effect of the parameter $k$ on the performance of representative methods. See Fig. A2 for more results on the effect of the parameter $k$. Overall, the performance of all methods is robust to the choice of $k$.

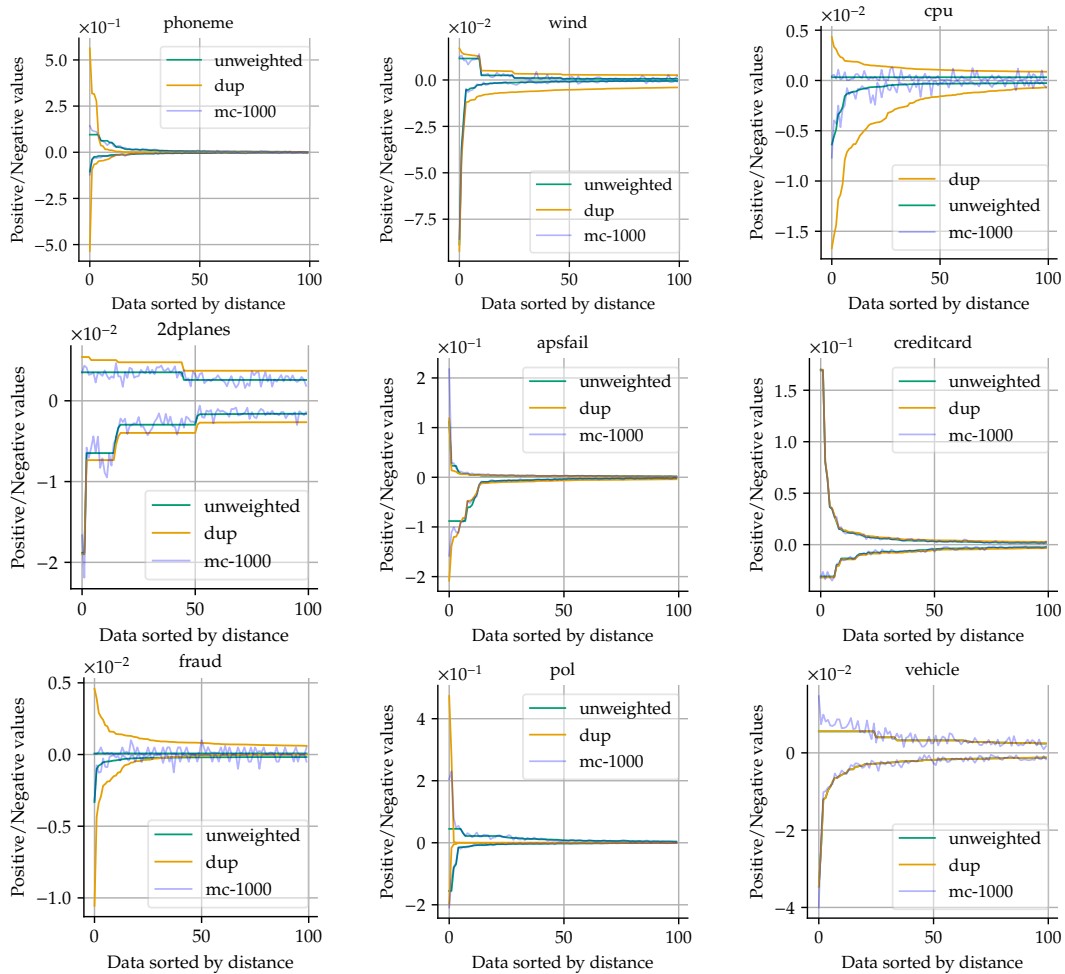

Figure A1: Visualization of the Shapley values of different methods concerning a random test point.

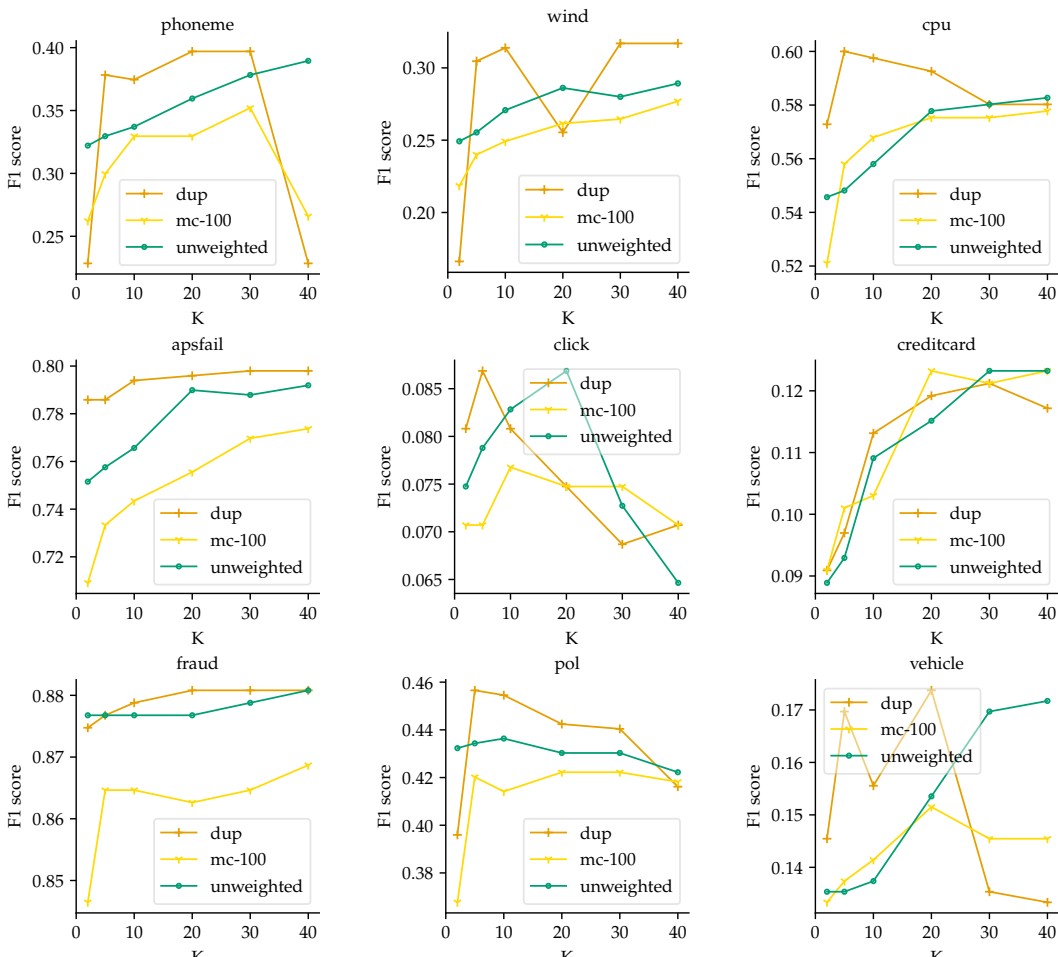

Figure A2: Effect of the parameter $k$ on the performance of different methods.

