# OpenReview forum: "Shapley-Based Data Valuation for Weighted $k$-Nearest Neighbors"
_NeurIPS.cc/2025/Conference — NeurIPS 2025 poster_

### Official Review · Reviewer_GPEG · 2025-06-23

**Clarity:** 4
**Significance:** 4
**Originality:** 4
**Rating:** 5
**Confidence:** 3

**Summary:**

The paper proposed an efficient algorithm to tackle computing data Shapley for weighted KNN models, extending previously established work on unweighted KNN-Shapley by Jia et al. The authors related the solution concept to Owen values and demonstrate although their algorithm might result in solution concept that are not exactly Shapley, it still satisfies some other rationality constraints.

**Questions:**

How would your work extend to regression settings? I suppose you might have to bin the differences between y_obs and y_tests in order to still use the counting trick? Do you have any other smarter ideas on this?

**Ethical Concerns:**

["NO or VERY MINOR ethics concerns only"]

**Final Justification:**

I don’t have further concerns. The rebuttal addressed most of my concerns. I remain very positive about this submission.

**Limitations:**

It's only works for classification problems but this is a very minor problem given the significant improvement in computational speed.

**Paper Formatting Concerns:**

No.

**Quality:**

4

**Strengths And Weaknesses:**

Strength: Many. The paper is beautifully written, the ideas are clear, contributions are clear, comparison to existing works are clear.

Weakness:
- Not a serious one but should you still call your method data Shapley in this case? Just like how Jia's other work on Data Banzhaf, isn't your work some kind of Data Owen that's trying to approximate Data Shapley for weighted KNN?

Very minor: in lline 140 I suppose you wanted to say you treat the weights as "integers" rather than "integrals"?

---

> ### Author Rebuttal · Authors · 2025-07-28
>
> We thank the reviewer for reviewing our paper, and recognizing our contributions.
> We also thank the reviewer for pointing out the typo in line 140.
> We answer your questions below.
>
>
> **Q1**:
> Not a serious one but should you still call your method data Shapley in this case? Just like how Jia's other work on Data Banzhaf, isn't your work some kind of Data Owen that's trying to approximate Data Shapley for weighted KNN?
>
> **A1**: While it is popular to create new names for proposed methods for easy recognition, we didn't make an effort to do so.
> Thank you for suggesting a candidate, and we are thinking of using "kNN-GroupShapley" if a name is needed.
>
>
> **Q2**:
> How would your work extend to regression settings? I suppose you might have to bin the differences between y_obs and y_tests in order to still use the counting trick? Do you have any other smarter ideas on this?
>
> **A2**: Extending to regression is exactly one of the possible future works we mentioned in the conclusion.
> If the reviewer is interested in this, we recommend taking a look at Section E.1 of [13], where they define a utility function for regression with unweighted kNN as the negative mean squared error, and develop a similar recurrence relation for the Shapley values.

---

> > ### Comment · Reviewer_GPEG · 2025-08-08
> >
> > Dear Authors,
> >
> > Thank you for the clarification. I think from a scientific point of view, calling your method Data Shapley is a bit untruthful given it is not really the classical Shapley solution concept. Nonetheless, the technical contribution of this paper is very good so I will not dwell on this point further. Please consider honouring Owen more...
> >
> > Otherwise I have no other comments and remain positive about this paper.
> >
> > Best,
> > Reviewer

---

> > > ### Author Response · Authors · 2025-08-08
> > >
> > > This is a good point. We will take it into consideration when naming the method.
> > >
> > > Thank you again for your time and effort in reviewing our paper.

---

### Official Review · Reviewer_WDF7 · 2025-06-26

**Clarity:** 2
**Significance:** 3
**Originality:** 3
**Rating:** 4
**Confidence:** 4

**Summary:**

To address the issue of lacking fast Shapley values calculation methods in weighted kNN algorithms, this paper proposes duplication-based kNN-SV (dup). The algorithm applies data duplication to convert weighted kNN into unweighted kNN, and exploits efficient algorithms to compute Shapley values. At the same time, it reduces the complexity to near-linear time in the original data size. Experimental results on 11 datasets validate the effectiveness of dup.

**Questions:**

Please refer to the weaknesses above.

**Ethical Concerns:**

["NO or VERY MINOR ethics concerns only"]

**Final Justification:**

After reading the authors' rebuttal. I have decided to maintain my current score.

**Limitations:**

YES

**Quality:**

2

**Strengths And Weaknesses:**

Strength：

1.This paper proposes a new algorithm duplication-based kNN-SV (dup) that employs a duplication-based technique to efficiently reduce the weighted case to the unweighted setting.

2.The proposed algorithm not only preserves the weighted characteristics of kNN but also retains the desirable axiomatic properties of Shapley values while enabling computationally efficient estimation.

3.The experimental evaluation more or less demonstrated the effectiveness of the computationally efficient algorithm dup in noisy label detection.

Weakness：

1. In section 4, the authors assume weights are integral. This is a severe restriction, as weights are typically numerical. The authors mention that assumptions could be removed through weight scaling, but do not provide the specific operation.

2.In section 7, the comparison of the algorithms is insufficient. To further verify the effectiveness of the proposed algorithm, the authors should incorporate additional comparison algorithms about noisy label detection.

3. In section 7, the experimemtal results present standard deviation values but lack the significance test. This leads to the inability to confirm whether the performance differences between algorithms are statistically significant.

4.  In section 7, the datasets used in the experiments are not sourced. This makes me confused about the reason why the authors chose these datasets.

---

> ### Author Rebuttal · Authors · 2025-07-28
>
> We thank the reviewer for reviewing our paper.
> We answer your questions below.
>
>
> **Q1**: In section 4, the authors assume weights are integral. This is a severe restriction, as weights are typically numerical. The authors mention that assumptions could be removed through weight scaling, but do not provide the specific operation.
>
> **A1**: We want to emphasize that the assumption of integral weights is made without loss of generality.
> We now explain how to do weight scaling in detail.
> Suppose the weights are real numbers $w_1, \ldots, w_n$.
> We can scale the weights to integers by multiplying a constant factor $c$ such that $c w_i$ is an integer for all $i$.
> Note that since real-valued weights are stored in the computer with finite precision, such an integer $c$ always exists, and the scaling does not change the prediction or the utility function of the $k$-NN classifier.
> In practice, we set $c$ to a sufficiently large numeric integer and perform rounding afterwards.
>
>
> **Q2**: In section 7, the comparison of the algorithms is insufficient. To further verify the effectiveness of the proposed algorithm, the authors should incorporate additional comparison algorithms about noisy label detection.
>
> **A2**: We understand the reviewer's concern.
> However, the main goal of this paper is not to claim that kNN-Shapley is the best approach for noisy label detection.
> Instead, we aim to show that the proposed weighted kNN-Shapley method is effective in comparison to the unweighted counterpart and other kNN-based approaches.
> We hope the reviewer will agree that we have made sufficient comparisons for this purpose.
>
>
> **Q3**: In section 7, the experimental results present standard deviation values but lack the significance test. This leads to the inability to confirm whether the performance differences between algorithms are statistically significant.
>
> **A3**: Following the advice of the reviewer, we added a significance test to the experimental results.
> We choose the Wilcoxon signed-rank test, which is a non-parametric test that does not assume a normal distribution of the F1 scores.
> Taking the F1 scores of each method over 11 datasets as input and using a significance level of 0.05, the performance of our weighted kNN-Shapley method is statistically significantly better than all baselines except for the unweighted kNN-Shapley method and the Monte Carlo sampling method with 1000 samples.
> However, if we slightly raise the significance level to 0.06, our method outperforms the unweighted kNN-Shapley method in the test.
> Note that it is not expected that our method can defeat the Monte Carlo sampling method with a large number of samples, because the latter is also based on weighted kNN-Shapley, but is much slower.
> We hope that this test helps to justify the effectiveness of our method.
>
>
>
> **Q4**: In section 7, the datasets used in the experiments are not sourced. This makes me confused about the reason why the authors chose these datasets.
>
> **A4**:
> We adopt most datasets that are used in the previous work [29,30] on kNN-Shapley.
> We added a few new datasets with a larger size (e.g., poker), which is a popular dataset available in UCI Machine Learning Repository.

---

### Official Review · Reviewer_B26m · 2025-07-02

**Clarity:** 4
**Significance:** 2
**Originality:** 2
**Rating:** 4
**Confidence:** 4

**Summary:**

This paper introduces a novel approach to computing Shapley values for weighted k-nearest neighbor (kNN) classifiers through a data duplication technique. While existing methods for computing exact Shapley values in kNN models are limited to unweighted cases or require computationally expensive approximations for weighted variants, the authors propose creating multiple copies of each data point proportional to its weight, effectively reducing the weighted kNN problem to an unweighted one.

**Questions:**

N/A

**Ethical Concerns:**

["NO or VERY MINOR ethics concerns only"]

**Final Justification:**

The technical contribution lies on the low end, but this won't be a reason to reject the paper. I appreciate the additional algorithm details and experiments. Overall, I think it's a nice paper on KNN-Shapley that is worth sharing with the NeurIPS community. Therefore, I recommend accept.

**Limitations:**

I decided to set my score as 3 given (1) the limited theoretical contribution, (2) insufficient discussion on algorithm details, and (3) missing important baselines from the experiments. Happy to raise the score if all the concerns are addressed during the rebuttal.

**Quality:**

3

**Strengths And Weaknesses:**

## Strengths
- Data valuation is an important problem for ML community and KNN-Shapley is a very popular approach.
- The paper is well-written and accessible, with comprehensive background material that facilitates understanding.

## Weaknesses
- Section 3 is still the preliminaries from prior works and should be merged into Section 2. Additionally, Theorem 1 should cite Wang and Jia's technical note in its title.
- Line 140 should use "integer" instead of "integral".
- I feel the theoretical contribution falls on the low end. For example, Proposition 2 appears to be a straightforward observation. Theorem 3 merely restates standard Shapley axioms with minor adaptation.
- Theorem 6's runtime analysis omits the dependency on K', with only a brief mention that "partial sum of the harmonic series can be accurately approximated by the Euler-Maclaurin formula [2], or retrieved from a pre-computed table." This is problematic because:
  - If approximation is used, it introduces an additional layer of error that should be characterized
  - If pre-computed tables are used, this significantly increases memory requirements
  - Algorithm 1 should explicitly include these details in the pseudocode and acknowledge any limitations.
- For experiments, the most relevant baseline—hard-label weighted kNN-Shapley by Wang et al. [1]—is omitted with insufficient justification. The authors claim it "fails to finish within 5 hours on small datasets," but after reviewing [1] I do not think this explanation is sufficient:
  - Runtime of hard-label WKNN-Shapley can be controlled by adjusting discretization bits for weights
  - [1] also offers a deterministic approximation algorithm with sub-quadratic complexity (Section 4.2 of [1])

[1] Wang, Jiachen T., Prateek Mittal, and Ruoxi Jia. "Efficient data shapley for weighted nearest neighbor algorithms." International Conference on Artificial Intelligence and Statistics. PMLR, 2024. https://arxiv.org/pdf/2401.11103

---

> ### Author Rebuttal · Authors · 2025-07-28
>
> We thank the reviewer for the thorough review, and the constructive suggestions on improving the presentation of the paper.
> We answer your questions below.
>
>
> **Q1**:
> I feel the theoretical contribution falls on the low end. For example, Proposition 2 appears to be a
> straightforward observation. Theorem 3 merely restates standard Shapley axioms with minor adaptation.
>
> **A1**: Our main idea about data duplication is indeed a simple one, but it is effective and has been overlooked by prior work.
> Other reviewers consider the simplicity of our approach as a strength with clear benefits (e.g., see Reviewer KFsJ).
> Besides, we propose techniques to avoid the computational burden caused by data duplication, and provide theoretical analysis of the difference from other approaches.
> We believe these contributions together establish a complete theoretical and practical foundation for our proposed approach.
>
>
>
> **Q2**:
> Theorem 6's runtime analysis omits the dependency on $K'$, with only a brief mention that "partial sum of the
> harmonic series can be accurately approximated by the Euler-Maclaurin formula [2], or retrieved from a precomputed table." This is problematic because:
> If approximation is used, it introduces an additional layer of error that should be characterized
> If pre-computed tables are used, this significantly increases memory requirements
> Algorithm 1 should explicitly include these details in the pseudocode and acknowledge any limitations.
>
> **A2**: We thank the reviewer for pointing out the missing details.
> Asymptotically, the Euler-Maclaurin formula $F(j)$ below converges to the harmonic series $H_j = \sum_{i=1}^{j} \frac{1}{i}$ as $j \to \infty$.
> $$
> F(j) = \ln(j) + \gamma + \frac{1}{2j} - \frac{1}{12j^2},
> $$
> where $\gamma \approx 0.5772156649$ is the Euler-Mascheroni constant.
> In practice, we adopt the following more accurate approximation scheme.
> When $j$ is a small constant (e.g., $j \le 1000$), we can afford to compute $H_j$ exactly.
> Otherwise, we use $F(j) - F(1000) + H_{1000}$ as an approximation (in $O(1)$ time where $H_{1000}$ is pre-computed) to eliminate the noticeable error when $j$ is relatively small.
> We verify the absolute error is at most $4.167 \times 10^{-8}$ for $j$ up to 1 million and continues to decrease as $j$ increases.
> We believe this approximation is sufficient for our purpose.
>
>
>
>
> **Q3**:
> For experiments, the most relevant baseline—hard-label weighted kNN-Shapley by Wang et al. [1]—is omitted
> with insufficient justification. The authors claim it "fails to finish within 5 hours on small datasets," but after
> reviewing [1] I do not think this explanation is sufficient:
> Runtime of hard-label WKNN-Shapley can be controlled by adjusting discretization bits for weights.
> [1] also offers a deterministic approximation algorithm with sub-quadratic complexity (Section 4.2 of [1]).
>
> **A3**: In our experiments, we use the default discretization bits ($n_{bits}=3$) for this baseline, which only enables at most 8 different weight values.
> We believe no significant saving in time can be expected when using fewer bits.
>
> Instead of resorting to the approximate version, we choose to compare with the exact version over small datasets.
> We set the data size to be 300 by randomly sampling from the original datasets.
> Very surprisingly, the hard-label WKNN-Shapley is only slightly better than random guessing, and is far behind the performance of the unweighted kNN-Shapley and our method.
> To make sure this is not caused by bugs in the code, we further verify that its values are indeed consistent with those by Monte-Carlo sampling with a hard-label utility function.
>
> After careful inspection, we identify the key reason: a hard-label utility function is not suitable for the task of noisy label detection.
> Unlike the soft-label utility, it is unable to capture the fine-grained contribution of a data point.
> It requires the noisy point to be a game changer for the prediction of its neighbors in order to be considered harmful, which is not the case for most mislabeled points.
> A mislabeled point is often surrounded by well-labeled points, and its ability to change the prediction of its neighbors is often negligible.
> Note that resorting to a hard-label utility function is the key modification that enables the DP algorithm in [1] to work.
>
> We thank the reviewer for suggesting further comparison with this baseline, which leads to insightful observations.
> We will add this comparison in the revised version of the paper.
> We hope the reviewer will consider raising the score.

---

> > ### Comment · Reviewer_B26m · 2025-08-05
> >
> > Thanks to the author for the detailed response.
> >
> > Regarding your analysis of hard-label WKNN-Shapley, I agree that mislabeled points rarely significantly alter KNN's hard-label prediction. However, I believe the issue is more complex. The Shapley value inherently considers all possible subsets when computing marginal contributions, so prediction-flipping events *are* captured in the computation. The question is whether these signals are sufficiently strong. The more fundamental challenge is that WKNN-Shapley's effectiveness depends on the absolute magnitudes of distance-based weights, not merely relative rankings like unweighted KNN-Shapley. This creates a significant hyperparameter sensitivity problem, particularly with the kernel width parameter that controls how rapidly weights decay with distance. Finding the "just right" configuration is critical but difficult, making the weighted method unstable and less reliable for a task like noisy label detection compared to the simpler and more robust rank-based approach of unweighted kNN-Shapley.
> >
> > I think all my concerns have been addressed. I have also read other reviews, and I agree that this is an interesting work to share at NeurIPS. Please incorporate the suggested changes for the paper presentation. For the naming, I am not sure about KNN-GroupShapley since it sounds like group-level Shapley value. My suggestion is Dup-KNN-Shapley or DKNN-Shapley.
> >
> > Congrats for the great work!

---

> > > ### Author Response · Authors · 2025-08-06
> > >
> > > We thank the reviewer for the insightful comments.
> > > The comment about WKNN makes sense to us, and a right configuration is indeed crucial.
> > > We will incorporate more discussion about WKNN in the revised paper.
> > >
> > > DKNN-Shapley sounds good to us. Thank you for the suggestion!

---

### Official Review · Reviewer_KFsJ · 2025-07-17

**Clarity:** 4
**Significance:** 3
**Originality:** 3
**Rating:** 5
**Confidence:** 4

**Summary:**

This paper is building on previous work leveraging $k$-NN classifiers in order to reduce the computational burden of the Shapley value, in order to perform data valuation. Current $k$-NN based approaches consider either (1) un-weighted $k$-NN models (where neighbors are not penalised with respect to their distance to the target data sample); (2) or weighted $k$-NN models but with prohibitive quadratic computational cost.

This paper is an attempt to fill this gap by proposing a novel approach leveraging weighted $k$-NN models with almost linear (w.r.t. the full coalition size $n$) computational cost, i.e. of the order of $O(nlog(n))$. To that purpose, the authors are proposing the following contributions:
- a novel variant of weighted $k$-NN Shapley value by duplicating the training dataset, which is shown to meet the unweighted variant (Proposition 2);
- a thorough discussion of the aforementioned proposed framework compared to related techniques (e.g. Owen values and other $k$-NN Shapley-based data valuation techniques (Sections 4.1 & 4.2);
-  a scalable data valuation algorithm avoiding the naive additional computational burden related to data duplication;
- experiments showing the benefits of the proposed approach.

**Questions:**

See my concerns in the above section.

**Ethical Concerns:**

["NO or VERY MINOR ethics concerns only"]

**Limitations:**

Yes

**Quality:**

3

**Strengths And Weaknesses:**

**Strengths**
- The paper is very well written, clear and contributions well identified compared to past work.
- The discussion and comparison with related approaches (notably Owen and $k$-NN based) are thorough and insightful.
- The contribution is timely and of interest for the machine learning community.
- Support from both theoretical insights and experimental ones is adequate.
- The proposed approach is simple (I consider this as a strength) but its benefits are clear.

Overall, the paper looks solide and mature.

**Weaknesses**

- [Major] Could you discuss in further details the impact of the dimensionality $d$ compared to unweighted variants, on the scalability of the proposed approach?
- [Minor] $K$ is not defined in (3), I believe it is $k$?

---

> ### Author Rebuttal · Authors · 2025-07-28
>
> We thank the reviewer for reviewing our paper and recognizing our contributions.
> We answer your questions below.
>
> **Q1**: [Major]
> Could you discuss in further detail the impact of the dimensionality $d$ compared to unweighted variants, on the scalability of the proposed approach?
>
> **A1**:
> As far as we know, the dimensionality $d$ plays no special role in the comparison between the weighted and unweighted kNN-Shapley.
> It only affects the sorting stage, which is *required* for both the weighted and unweighted kNN-Shapley.
> When given a test point $z_{test}$, we need to compute the distance between $z_{test}$ and all the training points in $D$ before we can sort $D$ by non-decreasing distance to $z_{test}$.
> Thus, the time complexity of distance computation (e.g., Euclidean distance) is $O(d)$ for each pair of points, which leads to $O(dn)$ in total.
>
>
> **Q2**: [Minor]
> $K$ is not defined in (3), I believe it is $k$?
>
> **A2**: Yes, you are correct. We will fix the typo. Thanks.

---

### Decision · Program_Chairs · 2025-09-17

**Decision:**

Accept (poster)

**Comment:**

This work proposed a new variant of the Shapley value for k-nearest-neighbor classifiers that can be efficiently computed, improving the previous quadratic complexity to (near) linear complexity. The reviewers praised the presentation of the paper and the simplicity (and effectiveness) of the idea. The authors' response addressed most of the reviewers' questions and resulted in a uniformly positive recommendation, which I concur.

Some extra comments for the authors to consider incorporating:

- The abstract, unlike the introduction, did not make clear the proposed method is a variant of knn-Shapley. Thus, technically speaking, this work did not bridge the gap of the quadratic complexity of knn-Shapley.

- The paper should make clear that only **rational** weights are dealt with. The authors in the rebuttal mentioned that
  > Suppose the weights $w_i$ are real numbers. We can scale the weights to integers by multiplying a constant factor $c$ such that $c w_i$ is an integer for all i.

  This is incorrect: let $a\ne 0$ be rational and $b$ be irrational. We can easily prove that there does not exist $c\ne 0$ such that $ac$ and $bc$ are both rational.

  Of course, in practice numbers are represented in finite-precision, making them effectively rational. However, it is best to be clear about what is being dealt with in theory and then comment on the practical effectiveness.